# Advancing Management of Oral Lesion Patients with Epidermolysis Bullosa: In Vivo Evaluation with Optical Coherence Tomography of Ultrastructural Changes after Application of Cord Blood Platelet Gel and Laser Photobiomodulation

Alessio Gambino *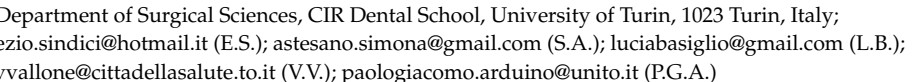, Ezio Sindici, Simona Astesano, Lucia Basiglio, Valeria Vallone and Paolo Giacomo Arduino

Department of Surgical Sciences, CIR Dental School, University of Turin, 1023 Turin, Italy;
ezio.sindici@hotmail.it (E.S.); astesano.simona@gmail.com (S.A.); luciabasiglio@gmail.com (L.B.);
vvallone@cittadellasalute.to.it (V.V.); paologiacomo.arduino@unito.it (P.G.A.)
* Correspondence: alessio.gambino@unito.it; Tel.: +39-0116331522

**Abstract: Background:** Inherited epidermolysis bullosa (EB) is a group of genetic disorders with skin fragility and blistering. The use of Cord Blood Platelet Gel (CBPG) in combination with laser photobiomodulation (PBM) leads to a reduction in lesions. The aim of this study is to evaluate clinical and morphometric changes with Optical Coherence Tomography (OCT) during GPC-PBM therapy. **Methods**: OCT scanning before the first session (T0), with relative measurement of the thicknesses of the epithelium (EP) and lamina propria (LP), and three consecutive sessions (once daily for 3 days) of CBPG and PBM applications were performed. A new OCT scan at the end of the three sessions (T1) and a week after (T2) were performed. All OCT scans were compared with the values of healthy reference tissues of the same site. **Results**: A statistically confirmed increase in EP thickness and a decrease in LP thickness with a progressive reduction in inflammatory content were highlighted. This case series did not have recurrences in the treated sites, or adverse reactions to therapy. **Conclusions:** This study shows the advantages of OCT monitoring in evaluating the effects of therapy at an ultrastructural level with a possibility of obtaining objective, precise, and repeatable measurements with an atraumatic device.

**Keywords:** epidermolysis bullosa (EB); dystrophic epidermolysis bullosa (DEB); optical coherence tomography (OCT); laser photobiomodulation (PBM); cord blood platelet gel (CBPG)

## 1. Introduction

Epidermolysis bullosa (EB) is a rare genetic disease that causes mucocutaneous fragility. It comprises a clinically and genetically heterogeneous group of disorders characterized by spontaneous or contact/friction-induced blistering. The disease is caused by defects in proteins implicated in dermal–epidermal adhesion. EB is classified into four types (simplex (EBS), junctional (JEB), dystrophic (DEB), Kindler syndrome (KEB)) and 30 subtypes. The subclassification considers clinical phenotypic features such as distribution (localized vs. generalized), relative severity of cutaneous and extracutaneous involvement, mode of transmission, and specific gene involved. At least 19 genes have been characterized and more than 1000 mutations identified, thus rendering diagnosis complex [1].

The estimated incidence of EB is 19.6 per 1 million live births and the prevalence is 11 cases per million inhabitants. By the age of 15, the cumulative risk of death is 62% in severe JEB and 8% in severe recessive DEB. Diagnosis is carried out by correlating clinical, histopathological microscopy, and immunohistochemistry features with mutational analysis of the genes implicated. The main laboratory tests are dedicated to immunofluorescence mapping and genetic tests, which help identify the altered or missing protein and the affected gene [2].

EB patients present perioral involvement such as microstomia and intraoral hard tissue involvement such as caries, enamel hypoplasia (localized or generalized), failure of eruption, occlusal abnormalities, dental crowding, and facial growth. The main intraoral soft tissue involvement includes oral ulcers, ankyloglossia, and vestibule obliteration and consequently, patients have difficulties during oral hygiene maneuvers. Even dentists and dental hygienists can also have difficulties during therapies given the limited space to open the mouth, especially for the posterior sector of the oral cavity.

In the absence of definitive curative treatment, multidisciplinary care is aimed at minimizing the risk of blistering, wound healing, symptom relief, and specific complications, the most feared of which is squamous cell carcinoma [3].

Previous studies have already demonstrated the effectiveness and safety of a cord blood-derived platelet gel (CBPG) with photobiomodulation therapy (PBM) for the treatment of oral EB lesions [4]. The goals of this combined protocol were pain reduction, increased mouth opening, wound healing, and no side effects.

However, the evaluation of healing is still empirical based on clinical experience, which is not always objective, especially when the treated tissues do not heal completely. In recent years, the knowledge and use of Optical Coherence Tomography (OCT) for oral tissues have developed [5].

This is a non-invasive imaging technique that measures the amplitude of backscattered light generated by a light source as a function of depth and produces an optical scan of the epithelial microstructures, distinguishing what is healthy from unhealthy. The resolution of the images obtained with this technique is very close to that of a low-magnification optical microscope with a tissue penetration of 2 mm. It is possible to identify epithelium (EP) and lamina propria (LP) in a very similar way to traditional histological preparation [6].

OCT is proposed as a valid aid in the measurable evaluation of oral tissues before and after topical therapy [7].

The aim of this study is to analyze clinical and morphometric changes with OCT during combined GPC-PBM therapy in EB patients.

## 2. Materials and Methods

### 2.1. Patient Selection

The ethical review board of the "Azienda Ospedaliera Città della Salute e della Scienza of Turin", Turin, Italy, approved this study (protocol number 0089210_CS/585/09-2015).

EB patients with blisters and/or erosions localized on intraoral and perioral tissues, measuring no more than 2 cm in diameter, who had not applied topical treatments in the previous 30 days were selected. It was recommended to patients not to carry out topical therapies on the area to be treated. Uncooperative patients, those younger than 6 years of age, and those under systemic corticosteroid, immunomodulatory, or chemotherapy therapies for other comorbidities were excluded. A specific informed consent form on the protocol to be carried out was explained and then signed by the parents or legal representatives of the minor patients.

### 2.2. CBPG Storage

The CBPG was created according to the "Italian Cord Blood Platelet Gel Project" protocol [8]. Cord blood units were prepared at Cord Blood Bank–Immunohematology and Transfusional Medical Service, A.O.U. Città della Salute e della Scienza, Turin, Italy.

All units were negative in infectious marker screening for HIV, HCV, HBV, syphilis, and HTLV, in compliance with standard criteria and procedures. Adult peripheral blood units were processed to obtain a CB platelet concentrate (CBPC) with a mean platelet count of 1000, range 800–1200 $\times 10^9$/L. The CBPC units were finally transferred into a storage bag and cryopreserved without a cryoprotectant in a mechanical freezer at $-80\,^\circ$C. At the time of clinical use, the CBPC units were thawed at 37 $^\circ$C in a water bath and the formation of a platelet gel was activated by the aseptic addition of 1/3 vol. of 10% calcium gluconate.

Activated CBPG units were used within 6 h of activation. Subsequently, CBPG was applied to oral lesions with a brush or syringe without a needle (Figure 1).

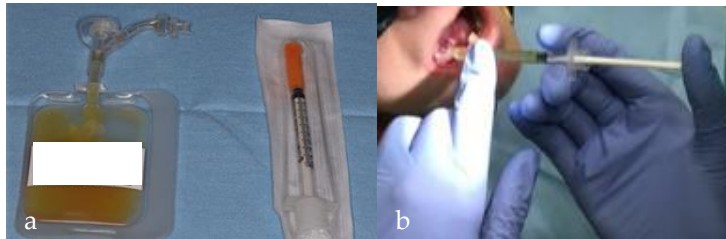

**Figure 1.** (**a**) A unit of CBPG and syringe; (**b**) topical application of CBPG on an oral lesion.

### 2.3. Laser Device for PBM

Immediately after the application of CBPG on the lesion site, PBM laser treatment was performed. Each lesion was irradiated using light with a wavelength of 645 nm, emitted by a gallium–aluminium–arsenide (Ga-Al-As) diode laser at 980/645 nm. A handpiece capable of emitting a collimated beam with a spot area of 0.5 cm$^2$ was used. The laser was operated in continuous mode, with a point-to-point technique, positioning the handpiece perpendicularly to the lesion at a distance of approximately 2 mm (Figure 2). Each lesion was irradiated with a power density of 500 mW/cm$^2$ and a fluence of 8 J/cm$^2$ for 16 s, with a power setting of 250 mW, delivering a total energy of 4 J per application point. The PBM parameters used are summarized in Table 1.

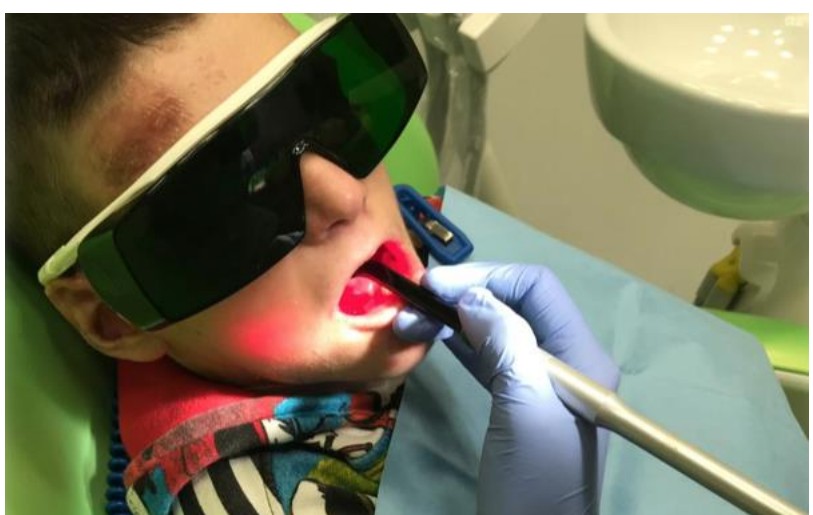

**Figure 2.** Laser PMB treatment after application of CBPG.

**Table 1.** Parameters for laser PBM protocol used.

| | |
|---|---|
| Manufacturer | DMT s.r.l. Lissone (MB) Italy |
| Model identifier | Raffaello 980 |
| Year produced | 2014 |
| Number and type of emitters | Diode laser |
| Wavelength | 645 nm |
| Beam spot size at target | 0.5 cm$^2$ |
| Pulse mode | CW |
| Irradiance at target | 250 mW |
| Pulse peak irradiance | 500 mW/cm$^2$ |

**Table 1.** *Cont.*

| | |
|---|---|
| Exposure duration | 16 s |
| Radiation exposure | 8.0 J/cm$^2$ |
| Number of points irradiated | 1 |
| Area irradiated | 0.5 cm$^2$ |
| Application technique | point by point |
| Number of treatment sessions | 3 |
| Total radiant energy | 4.0 J |

*2.4. OCT Device*

A variant of an OCT device, made for dermatological purposes, was used (SS-OCT System Vivosight® Michelson Diagnostics Ltd Maidstone, Kent, UK). The same company has created a prototype probe usable for intraoral examination (OCT endoscopic variant, version 2.1) [9].

Clinical photographs of the lesions were taken before and after treatment. Width variations within the EP and LP layers were standardized by the numerical difference of both layers as compared to healthy tissue.

To reduce the risk of bias related to the intraoral variability of the patients, OCT scan measurements of healthy tissues from archival images were taken into consideration as the gold standard. The timing of operative protocol sessions proceeded as follows: OCT scanning, before the first session, directly on the lesions (T0), with relative measurement of the thicknesses of EP and LP; 3 consecutive sessions (once daily for 3 days) of CBPG and PBM applications; new OCT scan at the end of 3 sessions (T1) and a week after (T2). Finally, all OCT scans were compared with the values of the healthy reference tissues of the same site.

*2.5. Statistical Analysis*

The paired Student *t*-test was conducted to evaluate the changes in EP and LP amplitude from the beginning to the end of the treatment and compare them with the changes in the healthy reference tissue. Statistical analysis was performed using SAS ver. 9.3, and a 2-tailed *p*-value less than 0.01 was considered statistically significant.

**3. Results**

Twenty D-EB patients with oral lesions were selected in the period from September 2022 to July 2023. The mean age of the patients was 11.5 years, 12 male and 8 female.

All patients were affected by D-EB. At the end of the combined therapeutic protocol, overall, in erosive lesions, clinical healing of the treated sites was observed.

Considering OCT scans of healthy tissue as a reference (Figure 3a), at T0, we observe the complete absence of EP and an increase in LP due to an inflammatory reaction (Figure 3b).

Due to the absence of EP at T0, only LP measurements were considered.

At T1, a large part of the lesions presented a return of EP and a mild decrease in LP. (Figure 3c). Specifically, LP after the first session of therapy (T0–T1) had a mean value of 0.83 (±0.03) mm, in contrast to a mean width of 0.76 (±0.03) mm registered before therapy: this could represent the progressive re-epithelization of the site with a reduction in inflammation only 24 h after treatment.

At T2, an increasing value of EP and LP and complete healing of the tissue were observed (Figure 3d).

On the other hand, EP experienced an increase after treatment (T1–T2), from a mean width of 0.1250 (±0.03) mm to 0.2210 (±0.03) mm, values substantially comparable to healthy ones (Table 2). A paired *t*-test revealed a two-tailed *p*-value < 0.001 (95% CI: −0.129;

−0.062), suggesting a statistically significant increase in EP width from T1 to T2 and LP from T0 to T2 (95% CI: −0.039; −0.108) (Table 3).

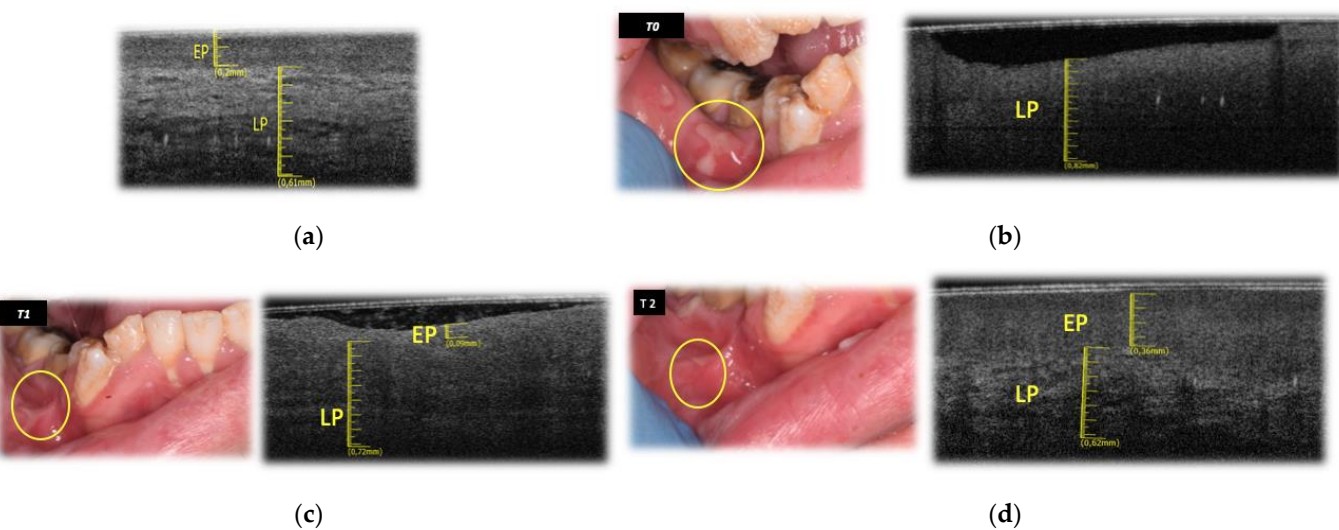

**Figure 3.** (**a**) Healthy tissue scan; (**b**) clinical images and OCT scan of erosion on the alveolar mucosa at T0; (**c**) clinical images and OCT scan of erosion on the alveolar mucosa at T1; (**d**) clinical images and OCT scan of after treatment on the alveolar mucosa at T2.

**Table 2.** Statistical parameter measurements from OCT scans before and after PBM-CPG treatment.

| Statistical Parameters | EP T1 | EP T2 | LP T0 | LP T2 |
|---|---|---|---|---|
| Sample size | 20 | 20 | 20 | 20 |
| Mean | 0.1250 | 0.2210 | 0.8385 | 0.7645 |
| SD | 0.5880 | 0.6206 | 0.0715 | 0.0574 |
| SEM | 0.0131 | 0.0138 | 0.0160 | 0.0128 |

SD (standard deviation), SEM (standard error of mean), EP (epithelium), LP (lamina propria).

**Table 3.** Differential paired *t*-test values considering $p < 0.01$.

| *t*-Test | Mean | SD | SEM | *p*-Value (95% C.I.) |
|---|---|---|---|---|
| EP T1—EP T2 | −0.0960 | 0.0715 | 0.0160 | <0.01 −0.1294; −0.0625 |
| LP T0—LP T2 | 0.0740 | 0.0743 | 0.0166 | <0.01 0.0392; 0.1087 |

SD (standard deviation), SEM (standard error of mean), EP (epithelium), LP (lamina propria).

The comparison between the values from the last session with the healthy tissue highlighted a statistical difference only for LP, while for EP, it was not significant ($p = 0.05$), where the statistical significance was set with a *p*-value < 0.01 (Table 4).

**Table 4.** Statistical parameter measurements from OCT scans after PBM-CPG treatment compared to healthy tissue.

| *t*-Test | Mean | SD | SEM | *p*-Value (95% C.I.) |
|---|---|---|---|---|
| EP T2—HT | −0.0230 | 0.0616 | 0.0137 | =0.05 (−0.0518; 0.0058) |
| LP T2—HT | −0.0645 | 0.0574 | 0.0128 | <0.01 (0.0913; 0.0376) |

SD (standard deviation), SEM (standard error of mean), EP (epithelium), LP (lamina propria) HT (healthy tissue).

These data indicate newly formed EP not different from healthy EP in its ultrastructure, and its reappearance within a few hours is proof of this.

LP shows a different increase compared to that of healthy tissues and, although very similar in its thickness, it is possible to identify an increase in collagen and fibroblast content, which is typical of fibrotic tissue healing in EB patients.

## 4. Discussion

Although blisters and erosions are the most common oral features of EB, there are a few studies published about therapeutic approaches for oral lesions. In 2001, Marini and Vecchietti reported that sucralfate suspension reduced the development and duration of oral mucosal blisters and ulcers with satisfactory results [10]. In addition to these strategies, mouthwashes and oral gels aimed at managing mucositis and oral lesions are commonly prescribed to patients with EB. The availability of these mouthwashes will vary between countries, and the clinical effectiveness will also vary among patients [11].

In recent years, a few clinical studies for the management of oral ulcerations in EB patients have been carried out. Firstly, the effectiveness of CBPG and PBM in reducing intraoral discomfort from ulcerations was tested in patients with dystrophic EB, and they were also effective in the long term [12]. In 2018, another pilot split-mouth study assessed the efficacy of CBPG, with or without the use of PBMT, in this type of disease. It was confirmed that the use of the combined protocol was superior to the use of CBPG alone [13]. For this reason, in this study, the double CBPG-PMB protocol was used exclusively: the stimulation of tissue healing of erosions is greater if laser-assisted activation occurs. In another clinical study in 2020, this type of topical treatment was tested over a 3-day treatment period: one application daily on long-standing symptomatic oral lesions of seven patients with dystrophic EB. The pain and clinical size of lesions improved from the first day of treatment [13].

A review from 2018 highlights that, given the specific difficulties in healing skin wounds in EB (epidermolysis bullosa) patients compared to many other morbid conditions, it is not always necessary to aim for complete closure of the lesions. In fact, even an improvement in the clinical situation can lead to a significant change in the patient's management and quality of life [14].

Our experience with mucosal lesions suggests clinical healing comparable to that of healthy tissues. Therefore, once the protocol was standardized, there was a need for an objective evaluation of the ultrastructural similarity of the healed tissues to healthy ones, using an atraumatic methodology to avoid the formation of blisters and erosions typical of EB patients.

The imaging of structural modifications of underlying tissues is important for monitoring wound healing. OCT has potential applications in evaluating the therapeutic efficacy of healing and characterizing scars [15].

Despite the various device-dependent limitations, OCT is currently one of few tools that can obtain tissue information in an atraumatic way.

In a 2008 study, it was observed that OCT (Optical Coherence Tomography) was able to detect inflammation, early re-epithelialization, and resorption of the collagen scaffold. These findings suggested the potential of OCT for non-invasive and high-resolution monitoring of assisted wound healing in vivo, longitudinally, and instantaneously [16]. OCT was able to provide stable patterns that delineate the various conditions of the oral mucosa. The distinction between oral cancer and healthy mucosa is the most significant achievement that OCT has been able to accomplish [17]. This is particularly important given that among the worst sequelae in EB (epidermolysis bullosa) patients is the development of oral cancer, in addition to basal cell carcinoma. The OCT patterns describing oral autoimmune bullous diseases [18] differ markedly from those of EB patients, which appear to be more limited and have less involvement of the basement membrane. The OCT pattern of erosions in EB patients, in fact, at T0, appears very similar to erosive lesions of oral potential malignant disorders, with a lamina propria (LP) layer increased in thickness and an absence of epithelium (EP) compared to healthy tissue [19]. At T1, however, after the first session of treatment, we noticed an OCT pattern very close to that of traumatic ulcers [20]. In fact, the

traumatic ulcers presented an EP of reduced thickness, with an irregular if not disrupted surface. The LP appeared to preserve its reflexivity with a small increase compared to the healthy scan. Comparing these results with the previously mentioned study [7] in patients affected by Oral Lichen Planus treated with PBM (photobiomodulation) without CBPG (Cord Blood Platelet Gel), it is possible to conclude that in EB patients, healing seems to be faster but with an ultrastructural pattern and related measurements more similar to healthy tissue. This confirms the efficacy of this type of therapy. One of the major limitations of this study is the poor availability of CBPG, currently available in hospitals that adhere to the "Italian Cord Blood Platelet Gel Project" [8]. On the other hand, the poor diffusion of OCT with a dedicated intraoral probe limits its scope of action for monitoring, not only clinically, the healing of lesions before and after therapy.

**5. Conclusions**

To date, this is the first report evaluating the healing of oral tissues in EB patients undergoing topical therapy through the in vivo use of OCT devices. Although it is not possible to draw definitive conclusions due to the limited number of patients treated with the combined CBPG–PBM protocol, encouraging results have been obtained regarding intraoral lesions in terms of pain control, promotion of improvement, and complete healing of oral tissue in EB (epidermolysis bullosa) patients. This preliminary clinical experience in EB patients highlighted the usefulness of OCT (Optical Coherence Tomography) in evaluating the effects of therapy using an atraumatic method and at an ultrastructural level, with the potential to obtain objective, precise, and repeatable measurements. It is necessary to continue with further studies that will also address the recurrence of lesions in the treated areas during a longer follow-up.

**Author Contributions:** Conceptualization, A.G.; methodology, A.G. and E.S.; software, P.G.A.; validation, V.V.; formal analysis, A.G. and S.A.; investigation, L.B.; resources, L.B.; data curation, E.S.; writing—original draft preparation, A.G.; writing—review and editing, P.G.A.; supervision, V.V.; funding acquisition, A.G. All authors have read and agreed to the published version of the manuscript.

**Funding:** This research was funded by Grant for Internationalizzation GAMA_GFI_22-01F-University of Turin about fundamental research for collaborative research projects with international partners within the scope of the university three-year Programme 2021-2023 (Board of Directors resolution no. 8/2022/V/31 of 28 July 2022).

**Institutional Review Board Statement:** The study was conducted in accordance with the Declaration of Helsinki, and the Ethics Review board of the "Azienda Ospedaliera Città della Salute e della Scienza of Turin", Turin, Italy, approved the study (protocol number 0089210_CS/585/09-2015).

**Informed Consent Statement:** Informed consent was obtained from all subjects involved in the study.

**Data Availability Statement:** The original contributions presented in the study are included in the article, further inquiries can be directed to the corresponding author/s.

**Acknowledgments:** The authors are grateful to Jon Holmes—CEO of Vivosight®—Michelson Diagnostics Ltd., UK, and UCL Eastman Dental Institute, Department of Maxillofacial Medicine & Surgery London, UK—for the loan of the OCT device. Thanks to Diego Merati—CEO of DMT s.r.l., Lissone Milano, Italy—for the loan of the diode laser device. Thanks to Immunohematology and Transfusional Medical Service of OIRM Hospital, Città della Salute e della Scienza di Torino, Turin, Italy for the storage of CBPG.

**Conflicts of Interest:** The authors declare no conflicts of interest.

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
