# Peer review of "Advancing Management of Oral Lesion Patients with Epidermolysis Bullosa: In Vivo Evaluation with Optical Coherence Tomography of Ultrastructural Changes after Application of Cord Blood Platelet Gel and Laser Photobiomodulation"

_2673-6373, doi:10.3390/oral4040035_

Round 1
Reviewer 1 Report
Comments and Suggestions for Authors
Excellent work.
Line 43-
“We have perioral involment as micorstomia and different intraoral hard tissue”- kindly rephrase
Line 57- "On the 57
other hand we haven’t a precise evaluation of the recurrences of lesions." please check this line.
Line 103- "The same Company has set up a prototype 103
probe usable for intraoral examination (OCT endoscopic variant, version 2.1). C"- please provide if any link available to support your claim
Line 123- "The mean age of the patients was 11,5 years old , 12 male and 8 female." - change it to 11.5
Comments on the Quality of English Language
Very few areas with confusing the explanations, already suggested to rephrase.
Author Response
Comments 1: Line 43- “We have perioral involment as micorstomia and different intraoral hard tissue”- kindly rephrase
|
|||
Response 1: Thank you for pointing this out. We agree with this comment. Therefore, we have rephrased the sentence in Line 49- “EB patients presenting perioral involment as micorstomia and different intraoral hard tissue”
|
|||
Comments 2: Line 57- "On the other hand we haven’t a precise evaluation of the recurrences of lesions." please check this line.
|
|||
Response 2: Agree. We have deleted the sentence(line 66).
Comments 3: Line 103- "The same company has prepared a prototype of probe usable for intraoral examination (endoscopic OCT variant, version 2.1). “ - indicate if there is a link to support your request.
Response 3: Agree. We added a citation of a study using the same device as this study [9] (line 148).
Comments 4: Line 123- "The average age of the patients was 11.5 years, 12 males and 8 females." - change it to 11.5
Response 4: Thank you. We have changed as suggested (line 170)
|

Reviewer 2 Report
Comments and Suggestions for Authors
Authors, it is worth continuing research on the use of a tool such as OCT in the prevention and treatment of various disorders in the oral cavity.
According to the reviewer, the article is concise, contains all the necessary information in all parts of the article; introduction, materials and methods, as well as in the discussion. There is no unnecessary content in the article - unnecessary increase in the volume of the article. Results are interesting both for clinicians - combination of CBPG and PMB therapy, and for scientists and clinicians - the effectiveness of OCT as a non-invasive, objective tool for assessing the condition of tissues - healthy tissues and after subsequent stages of treatment.
The reviewer fully agrees with the content of the summary.
Author Response
Thank you very much for taking the time to review this manuscript and for fully agreeing with its content.

Reviewer 3 Report
Comments and Suggestions for Authors
It is a relevant and well-written manuscript on a critical clinical disease that deserves to be published
Author Response

(The authors gave the same response as above.)

Reviewer 4 Report
Comments and Suggestions for Authors
The study aims to evaluate clinical and morphometric changes with Optical Choerence Tomography (OCT) during Cord Blood Platelet Gel and photobiomodulation (PBM) therapy for patients with Inherited epidermolysis bullosa (EB).
The study is interesting, well-written, and well-structured. Some observed issues need to be considered;
- The parameters of the PBM should be described in detail and in a table as recommended in the literature; please check this paper; “Hamblin MR. How to Write a Good Photobiomodulation Article. Photobiomodul Photomed Laser Surg. 2019”.
- The description of the used technique of the CBPG and PBM is not clear in the methods section, it would be better to improve the description by supporting that with some clinical photos and photos of the preparation procedures. This would make the manuscript more informative to the reader and more reproducible.
- It is recommended to add a paragraph at the end of the discussion section, where the authors describe the study's limitations according to this experience for better interpretation of the results and for other researchers to consider them in future studies.
- The conclusion section should be improved by decreasing the level of confirmation since the study is only a case series.
Comments on the Quality of English Language- Please avoid first-person citations and phrases such as we, I, our studies, etc. This comment is noticed in the whole manuscript including the introduction section. Please revise.
- Some sentences were provided without a subject (in particular in the conclusion section). Please revise.
Author Response
Comments 1: - The parameters of the PBM should be described in detail and in a table as recommended in the literature; please check this paper; “Hamblin MR. How to Write a Good Photobiomodulation Article. Photobiomodul Photomed Laser Surg. 2019”.
|
Response 1: Thank you for pointing this out. We agree with this comment. Therefore, we have modified the protocol PBM description (lines 113-122) and added a table (Table 1, lines 126-139) with the parameters used for the laser PBM as suggested and reported in the article of Hamblin MR, 2019.
|
Comments 2: The description of the used technique of the CBPG and PBM is not clear in the methods section, it would be better to improve the description by supporting that with some clinical photos and photos of the preparation procedures. This would make the manuscript more informative to the reader and more reproducible.
|
Response 2: Agree. We have, accordingly, revised and modified method section to emphasize this point. In particular we added the description of CBPG storage (lines 98-108) and added yet some photos to better support the protocol description (Figure 1a, 1b, 2).
Comments 3: It is recommended to add a paragraph at the end of the discussion section, where the authors describe the study's limitations according to this experience for better interpretation of the results and for other researchers to consider them in future studies.
Response 3: Thank you for this comment. We added at the end of discussion section the limitations of the study (lines 284-289).
Comments 4: The conclusion section should be improved by decreasing the level of confirmation since the study is only a case series.
Response 4: Thank you. We changed the conclusion section as suggest (lines 293-301).
|
4. Response to Comments on the Quality of English Language
|
Point 1: Please avoid first-person citations and phrases such as we, I, our studies, etc. This comment is noticed in the whole manuscript including the introduction section. Please revise.
|
Response 1: Thank you. We reviewed and modified the first person in the text (lines 66, 217, 218, 226).
Point 2: Some sentences were provided without a subject (in particular in the conclusion section). Please revise. Response 2: Thank you. We reviewed and modified the sentences without a subject in particular in the conclusion.
|

Round 2
Reviewer 4 Report
Comments and Suggestions for Authors
The manuscript is improved after the revision.
Comments on the Quality of English LanguageStill, some grammatical errors should be revised. There are many sentences without a subject.
Author Response
Thank you very much for taking the time to review this manuscript. Please find the detailed responses point by point below and the corresponding revisions/corrections highlighted/in yellow and blue track changes in the re-submitted files.
|
|
Response to Comments on the Quality of English Language
|
Point 1: Still, some grammatical errors should be revised. There are many sentences without a subject. Thank you. We reviewed and modified the sentences correcting grammatical errors and inserting the subject where it was missing. |
